# In Vitro Modeling as a Tool for Testing Therapeutics for Spinal Muscular Atrophy and IGHMBP2-Related Disorders

**DOI:** 10.3390/biology12060867

**Published:** 2023-06-16

**Authors:** Julieth Andrea Sierra-Delgado, Shrestha Sinha-Ray, Abuzar Kaleem, Meysam Ganjibakhsh, Mohini Parvate, Samantha Powers, Xiaojin Zhang, Shibi Likhite, Kathrin Meyer

**Affiliations:** 1The Research Institute at Nationwide Children’s Hospital, Columbus, OH 43205, USA; 2College of Medicine, The Ohio State University, Columbus, OH 43205, USA

**Keywords:** SMA, IGHMBP2, SMARD1, CMT2S, SMN, in vitro modeling

## Abstract

**Simple Summary:**

Spinal Muscular Atrophy (SMA) is a genetic disease that can cause infant mortality. It is typically caused by mutations in the SMN1 gene. On the other hand, mutations in the IGHMBP2 gene can lead to a range of diseases, including the rare form of SMA known as SMARD1, as well as Charcot–Marie–Tooth 2S (CMT2S). In this study, we developed a patient-derived in vitro model system to generate induced neurons and explore disease pathogenesis and test the response to gene therapies. The generated induced neurons from SMA and SMARD1/CMT2S patient cell lines were then treated with the clinical gene therapies AAV9.SMN (Zolgensma) for SMA and AAV9.IGHMBP2 for IGHMBP2-related disorders. We found that the SMA neurons had morphological defects that partially responded to treatment with AAV9.SMN, while the SMARD1/CMT2S neurons showed similar improvement after the restoration of IGHMBP2. The model also helped us identify whether an unclassified mutation was disease-causing in a patient with suspected SMARD1/CMT2S. These findings could help improve our understanding of SMA and SMARD1/CMT2S, as well as aid in the development of new treatments for these diseases.

**Abstract:**

Spinal Muscular Atrophy (SMA) is the leading genetic cause of infant mortality. The most common form of SMA is caused by mutations in the SMN1 gene, located on 5q (SMA). On the other hand, mutations in IGHMBP2 lead to a large disease spectrum with no clear genotype–phenotype correlation, which includes Spinal Muscular Atrophy with Muscular Distress type 1 (SMARD1), an extremely rare form of SMA, and Charcot–Marie–Tooth 2S (CMT2S). We optimized a patient-derived in vitro model system that allows us to expand research on disease pathogenesis and gene function, as well as test the response to the AAV gene therapies we have translated to the clinic. We generated and characterized induced neurons (iN) from SMA and SMARD1/CMT2S patient cell lines. After establishing the lines, we treated the generated neurons with AAV9-mediated gene therapy (AAV9.SMN (Zolgensma) for SMA and AAV9.IGHMBP2 for IGHMBP2 disorders (NCT05152823)) to evaluate the response to treatment. The iNs of both diseases show a characteristic short neurite length and defects in neuronal conversion, which have been reported in the literature before with iPSC modeling. SMA iNs respond to treatment with AAV9.SMN in vitro, showing a partial rescue of the morphology phenotype. For SMARD1/CMT2S iNs, we were able to observe an improvement in the neurite length of neurons after the restoration of IGHMBP2 in all disease cell lines, albeit to a variable extent, with some lines showing better responses to treatment than others. Moreover, this protocol allowed us to classify a variant of uncertain significance on IGHMBP2 on a suspected SMARD1/CMT2S patient. This study will further the understanding of SMA, and SMARD1/CMT2S disease in particular, in the context of variable patient mutations, and might further the development of new treatments, which are urgently needed.

## 1. Introduction

Spinal Muscular Atrophy (SMA) is a rare genetic disorder, with an incidence of 1 in 5000–10,000 live births, that affects the motor neurons in the spinal cord, leading to progressive muscle weakness and atrophy. The most common form of SMA is caused by autosomal recessive mutations in the Survival Motor Neuron 1 (SMN1) gene, located on 5q13.2 (SMA) [1]. The SMN1 gene encodes for the survival motor neuron (SMN), a deficiency in which leads to a loss of motor neurons in the spinal cord, causing weakness and the wasting of skeletal muscles [2,3,4]. SMA is divided into five subtypes (0–4) based on the age at onset, milestone achievement and the number of copies of SMN2. SMN2 is a highly homologous copy of the SMN1 gene, is found in humans and is located close to the centromere of the same chromosome. SMN2 is identical to SMN1, except for a cytosine (C)-to-thymine (T) substitution in exon 7 that leads to the alternative pre-mRNA splicing of exon 7. This results in the production of the majority (~90%) of short, non-functional SMN2 transcripts [2,4] and approximately 10% of functional full-length SMN2 transcripts encoding the SMN protein [2]. In humans, the number of copies of the SMN2 gene varies for each individual, ranging from zero (about 10 to 15% of the population) to five or more [2,3]. Thus, SMN2 copy number is recognized as one of the main disease modifiers for SMA clinical severity. SMA is divided into five subtypes (0–4) based on the age at onset, milestone achievement and the number of copies of SMN2 (Table 1) [2,4].

Fortunately, there are multiple FDA-approved treatment options available for SMA patients today. Nusinersen, an antisense oligonucleotide [5], and Risdiplam, a small molecule [6], act as SMN2-splicing modifiers. However, both of the treatments require multiple administrations in patients [5,6]. As of May 2019, onasemnogene abeparvovec-xioi (Zolgensma) has been approved by the Food and Drug Administration (FDA) as the first systemically delivered Adeno-associated virus 9-based gene therapy with designated use for infants diagnosed with SMA type 1 [1,7]. It is given as a one-time administration and can easily cross the blood–brain barrier and target the central nervous system at all regions of the spinal cord [7]. 

An extremely rare form of SMA is Spinal Muscular Atrophy with Respiratory Distress type 1 or distal Spinal Muscular Atrophy type 1 (SMARD1/DSMA1), which is caused by a mutation in the IGHMBP2 gene, located on chromosome 11. This form of the disease makes up approximately 1% of early-onset SMA cases [8]. IGHMBP2 is a ubiquitously expressed gene that encodes for Immunoglobulin Mu DNA Binding Protein 2, an ATPase/helicase of the SF1 superfamily with poorly understood function [9]. Autosomal recessive mutations in IGHMBP2 cause a broad clinical spectrum characterized by the degeneration of α-motor neurons and ganglion cells, ranging from distal muscle weakness with fatal respiratory distress/failure (SMARD1) to milder motor neuropathies with sensory neuropathies and lesser respiratory involvement (CMT2S) [9,10,11,12,13,14,15]. Patients often display a mixed clinical phenotype, and their pathology can change as the disease progresses [9,10,11,12,13,14,15]. There is currently little evidence of a genotype–phenotype correlation, as patients with the same mutations in IGHMBP2 may present vastly different disease courses, including variable time of onset, phenotype and progression [14,15]. Though several other modifiers have been reported in mouse models [16], so far, ABT1 is the only confirmed disease modifier found in humans [17]. For IGHMBP2 restoration, we have previously developed a promising AAV9-based gene therapy, which is currently in Phase I/IIa clinical trials for IGHMBP2-related disorders (NCT05152823) [18].

When it comes to modeling the pathophysiology and potential treatments for SMA, IGHMBP2-related disorders, and other neuromuscular disorders, in vitro modeling systems allow for efficient and high-throughput screening of therapeutics in the context of real patients by utilizing patient-derived cells featuring a variety of disease-causing mutations [19,20,21]. Over the past two decades, researchers have been reprograming patient somatic cells into induced pluripotent stem cells (iPSCs), which can then be differentiated into central nervous system (CNS) cell types for modeling genetically inherited diseases, including SMA [22,23,24,25,26,27,28,29,30,31,32,33,34,35]. However, iPSC is not the only option for in vitro modeling [20,21,36]. Direct conversion techniques provide a favorable alternative for in vitro disease modeling without the need for iPSC generation while offering numerous additional benefits. One key advantage is that the direct re-programming process is less time-consuming, with most protocols taking between 7 and 14 days [20,21,36], while it can take weeks to months to just establish iPSC lines. Furthermore, the differentiation of patient iPSCs into disease-relevant cells such as neurons and astrocytes can take approximately 3–4 weeks [20], while disease-relevant cells can be obtained within a week or two using direct reprograming of patient somatic cells. Additionally, iPSCs are generated as clonal cell lines, while direct-conversion cell lines are generated from a mixture of somatic cells, which mitigates the impact of somatic cell mutations in cell lines [19,20]. Furthermore, there is mounting evidence that crucial aging-/disease-related epigenetic markers in somatic cells are lost during iPSC conversion, and maintained to a far greater degree during direct conversion [21,37,38,39,40,41]. Additionally, human iPSCs are likely to undergo chromosomal changes during both early and late passages [42,43], which is reduced in direct conversion methods, as the use of reprogramming factors is not needed, and cells are not subjected to multiple passaging, which are both risk factors for chromosomal instability [20,21,42,43,44]. Overall, compared with iPSC modeling, direct conversion lowers the risk of contamination, saves time, ensures a stable karyotype, and preserves natural epigenetic variation within patient cells lines, yielding a cell population that better reflects the natural biological state of the neuromuscular disorders [19,20,21].

In this work, our laboratory optimized a direct conversion method to generate neurons directly from fibroblasts using small molecules [36,44], to produce a novel patient-derived in vitro model system for SMA and SMARD1. These neurons show hallmark disease phenotypes previously observed in iPSC models, including altered neuronal morphology and reduced conversion efficiency. Excitingly, the majority of these phenotypes are significantly improved upon treatment with AAV9.SMN (Zolgensma) or AAV9.IGHMBP2. Additionally, with this model, we were able to characterize a variant of uncertain significance (VUS) for SMARD1/CMT2S according to the morphology and response to treatment. This direct conversion in vitro model could be a useful tool in the future to study disease pathology, understand modifiers of disease severity, and find new therapeutic targets.

## 2. Materials and Methods

### 2.1. Skin Fibroblasts

Human skin fibroblasts were obtained from Coriell Institute and from collaborators. Informed consent was obtained from all subjects before sample collection. Receipt of human samples was granted through Nationwide Children’s Hospital Institutional Review Board. The primary fibroblasts were maintained and expanded in DMEM GlutaMAX media (Gibco, Waltham, MA, USA) with 10% fetal bovine serum (FBS) and 1% Anti-anti (Gibco).

### 2.2. Direct Conversion of Fibroblasts to Induced Neurons

Patient and healthy fibroblasts were directly converted to neurons using small molecules, as described previously, with a few modifications [36,44]. Briefly, 12-well plates or 10 cm plates were coated with poly-D-lysine (50 μg/mL, Sigma, St Louis, MO, USA) in borate buffer for one hour at room temperature. Next, the plates were washed with Dulbecco’s phosphate-buffered saline (DPBS) (Gibco, Waltham, MA, USA) and coated with laminin (10 μg/mL) in DMEM/F12 (Gibco) at 37 °C for 2 h. Fibroblast cells, at a density of 75,000 for a 12-well and 800,000 for a 10 cm plate, were seeded with fibroblast culture medium for 1 day. The next day, the cells were transferred into neuronal induction medium (DMEM/F12: Neurobasal [1:1]) (Gibco) with 0.5% N-2 (Gibco), 1% B-27 (Gibco), cAMP (100 μM, Sigma), and bFGF-2 (20 ng/mL, Peprotech, Cranbury, NJ, USA) with the following chemicals: VPA (0.5 mM, Sigma), CHIR99021 (3 μM, Axon medchem), repsox (1 μM, Biovision, San Francisco, CA, USA), forskolin (10 μM, Tocris, Bristol, UK), SP600125 (10 μM, Sigma), GO6983 (5 μM, Sigma) and Y-27632 (5 μM, Sigma). Half of the medium containing the chemicals was changed after 3 days with fresh induction medium. On the fifth day, cells were switched to neuronal maturation medium (DMEM/F12: Neurobasal [1:1] with 0.5% N-2, 1% B-27, cAMP (100 μM), bFGF-2 (20 ng/mL), BDNF (20 ng/mL, Peprotech) and GDNF (20 ng/mL, Peprotech,) with the following chemicals: CHIR99021 (3 μM), forskolin (10 μM) and SP600125 (10 μM). The induced neurons were then fixed for immunofluorescence or pelleted for Western blot.

For AAV9 treatment, we used a modified version of a published protocol [45]. Briefly, cells were seeded at the same density as previously described. After seeding, cells were treated with neuraminidase (NA). After NA treatment, AAV9.SMN or AAV9.IGHMBP2 at a Multiplicity of Infection (MOI) of 100,000 K was added in the NIM. After adding the virus, iN conversion continued according to the previously described methodology. Importantly, for each conversion replicate, the control, treated and untreated lines were cultured side-by-side. This approach ensured consistency and allowed for direct comparisons between the different experimental conditions. 

On day 7, chemically induced neuronal cells were fixed with ice-cold 4% paraformaldehyde (Sigma-Aldrich) and 0.1% glutaraldehyde (Sigma-Aldrich, St. Louis, MO, USA) for 20 min, and blocked with ice-cold DPBS with 4% goat serum and 0.2% Triton for 1 h as described previously [44]. Primary antibodies against TUJ1 (Biolegend, San Diego, CA, USA), Map2 (Novus Biologics, Centennial, CO, USA), GABA (Novus Biologics) and vGLUT (Thermo Fisher, Waltham, MA, USA) were diluted in blocking solution. Incubation of the primary antibody was performed overnight at 4 °C. The following day, cells were washed 3 times in DPBS before the secondary antibody (Alexa Fluor, Thermo Fischer Scientific, Waltham, MA, USA) and DAPI (Thermo Fisher Scientific), diluted in blocking solution, were applied for 1 h at room temperature. After three washes with DPBS, the 12-well plates were imaged using an EVOS fluorescent microscope. The captured images were subsequently processed using Adobe Photoshop.

### 2.3. Neuronal Morphology Analysis

Processed images were used to perform neuronal morphology analysis. The neuron length, conversion rate and other phenotypes were calculated via the manual analysis of 12 randomly selected (20× magnification) fields for each line from three replicates by a blinded investigator. The neurite length was computed from the same set of images using the SNT plugin of the image-analyzing tool Fiji [46]. The parameters analyzed were total neurite length, percentage of neurons with neurites (% Tuj1-positive cells with neurites over total Tuj1-positive cells ×100), percentage of neurons without neurites (% Tuj1-positive cells without neurites over total Tuj1-positive cells ×100) and neuronal conversion rate, as previously described [47,48,49,50,51] (% Tuj1-positive cells over the total amount of initially seeded cells, as measured by DAPI staining).

### 2.4. Western Blot

Cell pellets were lysed with RIPA lysis buffer (Thermo Fisher) followed by sonication for 10 s. Protein was quantified via DC Protein Assay (Bio-Rad, Hercules, CA, USA) and 50 µg of cell lysate was loaded onto 4–12% BIS-Tris PAGE gel (Thermo Fisher Scientific) and run at 120 volts for 1 h. Protein was transferred onto a PDVF membrane (Bio-Rad) and blocked for 1 h at room temperature using Odyssey blocking buffer (Li-COR, Lincoln, NE, USA). SMN primary antibody (Millipore, Burlington, MA, USA) was incubated overnight at 4 °C. The next day, the membrane was washed three times with Tris-Buffered Saline and 1% Tween, incubated with 1/12,000 diluted Li-COR secondary antibodies for one hour at room temperature and washed again three times before imaging the blot using Odyssey DLx LICOR Imaging System. Then, the image was processed using Image Studio Lite, version 5.2.

### 2.5. Digital Droplet PCR

Briefly, RNA was extracted from iNs as previously described using the trizol extraction method (ref, my cell reports) and reverse-transcribed using the Revert Aid Kit (Thermo Fischer Scientific). Reaction mixtures were assembled using ddPCR Supermix for Probes No dUTP (Bio-Rad, Hercules, CA, USA), TaqMan primers and probes (final concentrations of 900 and 250 nM, respectively) for both vector-derived cDNA and YWHAZ (housekeeping gene) (Integrated DNA Technologies, Coralville, IA, USA), and a cDNA template (6 μL) at a final volume of 25 μL. Each reaction was then loaded into a well of a ddPCR 96-well plate (Bio-Rad), heat-sealed with foil (Bio-Rad), and then loaded on an Automated Droplet Generator (Bio-Rad, Hercules, CA, USA). After droplets were generated and transferred to a 96-well PCR plate, plates were heat-sealed with foil, and amplified to the end point on a Bio-Rad C1000 Touch Thermal Cycler (Bio-Rad, Hercules, CA, USA). The PCR plate was subsequently scanned on a QX200 droplet reader (Bio-Rad, Hercules, CA, USA) and the data were analyzed using QuantaSoft software v 1.7 (Bio-Rad, Hercules, CA, USA).

## 3. Results

### 3.1. Direct Conversion of Fibroblast Allows for Rapid Differentiation into Induced Neurons (iNs)

Our laboratory successfully adapted and optimized a previously established protocol for the direct differentiation of induced neurons (iNs) using small molecules [36,44]. Briefly, the skin fibroblasts were treated with neuronal induction medium containing a VCRFSGY cocktail for 5 days, followed by neuronal maturation medium with CFS for 2 days. The protocol lasted 7 days, after which we saw a shift from a fibroblast to a more neuronal morphology (Figure 1A). To further characterize the resulting iNs, we determined the expression of key markers of neuronal differentiation, mainly Tuj1 and Map2. Our findings confirmed that the resulting iNs did indeed express pan-neuronal markers (Figure 1B), indicating their successful differentiation into neuronal cells. Furthermore, our analysis revealed that the resulting culture was composed of a mixed population of neurons, with some cells expressing additional markers such as GABA, which are characteristic of inhibitory neurons and vGLUT1, markers for glutamatergic neurons (Figure 1B). However, these neurons did not possess other markers such as CHAT, which is typically associated with cholinergic neurons. These data indicate that our protocol was able to effectively convert fibroblasts into iNs in vitro.

### 3.2. Neurons of SMA Patients Show Abnormal Morphology In Vitro

To evaluate the morphological features of SMA iNs, we obtained two primary fibroblast lines, both with the same SMN1 mutation and SMN2 copy number (Table 2).

Along with the SMA lines, we also utilized skin fibroblasts from two healthy individuals as age-matched controls. All the fibroblasts were treated with the direct conversion protocol for the generation of iNs. Interestingly, Tuj1 staining revealed that iNs derived from both the SMA lines exhibited a modified neuronal morphology as compared to the iNs derived from the healthy controls (Figure 2A). Specifically, SMA iNs showed considerably shorter neurites (Figure 2B), a decreased percentage of Tuj1+ neurons with neurites (Figure 2C) and a high proportion of Tuj1+ soma lacking neurites (Figure 2D). Interestingly, only one of the affected lines (SMA-1) demonstrated altered conversion efficiency, defined as the proportion of Tuj1-positive cells over the total number of DAPI-stained cells (Figure 2E). While the control lines exhibited a conversion percentage of 60–80% (our standard for this protocol), SMA1 had a substantially lower efficacy of around 40%. As expected, the SMA iNs showed reduced levels of SMN when quantified via Western blot compared to the age-matched controls (Figure 2F and Appendix A).

Next, we decided to use the first FDA-approved systematic gene therapy for SMA, AAV9.SMN (Zolgensma), to evaluate the impact of restoring SMN protein in our patient iNs. The fibroblasts were treated with AAV9.SMN at the start (day 1) of the direct conversion protocol. The AAV9-treated SMA iNs were analyzed for the same parameters of neuronal morphology and conversion percentage. AAV9.SMN treatment significantly improved all the parameters in the treated cells (Figure 2A–E) as a result of the restoration of SMN levels in SMA iNs (Figure 2F and Appendix A). These data demonstrate the robustness and utility of our direct conversion system in recapitulating the SMA disease phenotypes and the therapeutic potential of AAV9.SMN in vitro.

### 3.3. AAV9.IGHMBP2 Treatment Rescues the Disease Phenotype of SMARD1/CMT2S iNs

We expanded our in vitro model system to include other genetic causes of SMA (non-5q), specifically those related to autosomal recessive mutations in IGHMBP2. IGHMBP2-related disorders span a wide spectrum of disease severity from an early-onset, more severe SMARD1 phenotype to a late-onset, less severe CMT2S phenotype. Unfortunately, there is no genotype–phenotype correlation in IGHMBP2-related disorders. In this study, we established skin-derived fibroblast lines from two SMARD1/CMT2S patients in the middle of the disease spectrum (Table 3). Both patients harbored compound heterozygous mutations in the IGHMBP2 gene, with both mutations confirmed as pathogenic [8].

iNs were generated from SMARD1/CMT2S patient lines as well as healthy controls, as described earlier. Similar to the earlier study, we also checked the effect of IGHMBP2 restoration on these lines by utilizing AAV9.IGHMBP2 treatment. AAV9.IGHMBP2 gene therapy is currently in Phase I/IIa clinical trials for IGHMBP2-related disorders at Nationwide Children’s Hospital, Columbus, OH (NCT05152823).

We compared the neuronal phenotype of the iNs of patients with IGHMBP2 mutations to the controls, and the effects of treatment with AAV9.IGHMBP2. Tuj1+ staining of the iNs demonstrated that the SMARD1/CMT2S patient lines also had altered neuronal morphology compared to the healthy iNs, which was restored upon treatment with AAV9.IGHMBP2 (Figure 3A). Further analysis indicates that the SMARD1/CMT2S lines had significantly shorter neurite lengths (Figure 3B) and a lower number of neurites per tuj1+ soma (Figure 3C), and subsequently, a higher percentage of Tuj1+ neurons without neurites (Figure 3D,E) compared to the controls. Importantly, treatment with AAV9.IGHMBP2 improved all the phenotypes in both lines (Figure 3A–E).

Interestingly, similarly to SMA, one of the SMARD1/CMT2S lines (PT2) showed a decreased neuron Tuj1+ yield (Figure 3E). However, unlike the other neurite-related parameters affected, the reduced conversion rate phenotype was not rescued upon IGHMBP2 restoration with gene therapy.

Finally, to confirm whether the improvement of these phenotypes was indeed the result of IGHMBP2 restoration due to AAV9 treatment, we determined the transgene expression in the iNs using reverse transcription ddPCR with primers and probes specific to vector-derived IGHMBP2. The untreated iNs did not express the transgene, and only very low numbers of positive droplets were observed in the regular background range, while treated iNs had a high expression of vector-derived IGHMBP2 transcript (Figure 3F), indicating the successful transduction of AAV9.IGHMBP2 in iNs (Figure 3).

### 3.4. iN Can Help Identify Variants of Uncertain Significance according to Phenotype and Response to Treatment

We obtained a primary cell line from a suspected SMARD1/CMT2S patient who had a compound heterozygous mutation, with one confirmed pathogenic mutation and an additional mutation in the IGHMBP2 gene classified as a variant of uncertain significance (VUS) based on bioinformatic tools at the time of sample acquisition (Table 4). VUS classification and study is important as it can impact patient care and treatment decisions, especially in diseases where treatments are available [52,53]. As the use of genetic testing continues to expand, the accurate classification of VUS becomes increasingly important for patient care and informed decision-making [53].

In order to better understand the functional impact of the VUS in IGHMBP2 function, we directly converted the skin-derived fibroblasts from this patient into induced neurons (iNs). We observed that the VUS-iNs exhibited an altered neuronal phenotype similar to earlier known SMARD1/CMT2S iNs, including shortened neurite length, a reduced percentage of neurons with neurites and a higher percentage of neurons without neurites when compared to the age-matched controls (Figure 4A–E).

We further determined whether the observed effect can be rescued via treatment with AAV9.IGHMBP2 in the VUS iNs. Surprisingly, the treated VUS iNs showed a significant rescue of all the neuronal morphology parameters, as well as the neuron conversion rate, following a similar pattern to that observed in earlier confirmed IGHMBP2 iNs (as shown in Figure 3). These results suggest that the VUS may be pathogenic and can contribute to the observed phenotype. Moreover, this is the first study reporting the neuronal phenotype of a VUS in IGHMBP2-related disorders and its rescue with AAV9-mediated IGHMBP2 expression.

## 4. Discussion

This study utilizes the direct conversion of skin-derived fibroblasts into iNs from patients with Spinal Muscular Atrophy (SMA) and IGHMBP2-related disorders (SMARD1/CMT2S). Our findings demonstrate that iNs from patients with SMA and IGHMBP2-related disorders exhibit morphological irregularities, which can be alleviated by restoring functional protein (SMN1 and IGHMBP2, respectively) through AAV9-mediated gene therapies. Furthermore, this approach can also aid in evaluating variants of uncertain significance (VUS) based on phenotype and response to gene therapy.

The direct conversion method employed in this study offers several benefits over conventional reprogramming techniques, as iPSC generation, maintenance and differentiation are time-consuming and require significant specialized resources [19,20,36]. In addition, iPSC generation involves clonal selection steps, which may result in variations between individual clones derived from the same patient. In our current protocol, induced neurons can be generated within seven days of adding a small-molecule cocktail from a primary fibroblast source. Immunocytochemical analysis confirmed that the resulting iNs expressed pan-neuronal markers and exhibited a mixed population of GABAergic and glutamatergic neurons, indicating successful differentiation into neuronal cells. This mix of GABAergic and glutamatergic neurons has been previously reported in other direct reprogramming protocols [49], with some cells expressing both vGlut1 and GABA simultaneously (Figure 1). In spite of the contrasting roles played by these neurotransmitters, some studies have shown the simultaneous release of glutamate and GABA from neurons located in the ventral tegmental area (VTA) and entopeduncular nucleus (EP) [54,55,56]. One of the significant advantages of our optimized protocol is its high yield of Tuj1+ neurons, which exceeds 60 percent for healthy control lines. This is a marked improvement over previous attempts to convert fibroblasts into neurons, which had a yield of Tuj1+ neurons of only 15–30 percent, and an improvement over the original protocol, which showed a ratio of 30% of Tuj1+ neurons in the culture [20,21,36]. The yield increase between our protocol and previously published results could be due modifications in the coating. Previous protocols have used gelatin-based coats; however, we opted for a dual coating approach using poly-lysine (PLL) and laminin (PLL/Lam). Research has demonstrated that dual coating (PLL/Lam) significantly enhances neuronal differentiation in rat iPSC models [57] and PC12 models [58]. This approach has shown superior results compared to single coatings such as gelatin and fibronectin. In fact, gelatin has been reported to be a poor substrate for neurite outgrowth or homogeneous differentiation in iPSC-based models [57]. On the other hand, Laminin has been shown to improve differentiation and neurite growth [59,60], particularly when used in conjunction with dual coatings [57]. This rate is consistent between experiments (Appendix A) and across lines, with the inter experiment conversion rate coefficient of variation being ~10% across all the lines studied. Importantly, this consistently higher conversion efficiency in healthy lines makes it easier to identify the defects in neuronal conversion in patient lines, if any.

Cells generated through direct conversion methods preserve more epigenetic markings, enabling further investigation of patient and disease variation [21]. This is particularly important in SMA research, where studying the impact of different mutations while preserving as many epigenetic marks as possible is crucial. While SMN2 copy number remains the most critical disease modifier in SMA, other disease modifier genes susceptible to epigenetic modification, such as PSL3, have been reported [2,3,4,61]. Preserving epigenetic markings is especially significant in the IGHMBP2 disease spectrum, where there is currently limited evidence of a correlation between genotype and phenotype, whereby patients with the same mutations may present vastly different disease courses [10,15]. So far, only one gene, ABT1 has been confirmed as an IGHMBP2 disease-modifying gene in humans [17]. As other disease modifier genes or epigenetic markers for IGHMBP2 disorders are currently unknown [2,9,17], modeling tools that preserve as many epigenetic markings as possible are essential. Our model also provides a unique opportunity to study disease differences in a rapid manner and evaluate the response to therapies that are already in use. Moreover, it could be a valuable tool to identify downstream therapeutic targets.

While the direct reprogramming method used in this study presents several strengths, it is essential to address the inherent limitations associated with this protocol. The most prominent limitation revolves around the predominantly generated mixed population of neurons, predominantly comprising glutamatergic and GABAergic neurons as opposed to cholinergic neurons or motor neurons (Figure 1B). However, in spite of this limitation, the results of this study demonstrate that the iNs generated through our protocol are capable of reproducing the results of previous investigations using induced pluripotent stem cells (iPSCs). Specifically, previous studies have reported a significant reduction in neurite length in motor neurons derived from patients with type I and type III SMA [23,32,33], a finding that is replicated in the SMA iNs generated in this study (Figure 2). It is noteworthy that earlier research utilizing iPSC lines from SMA patients showed a decreased capacity for neuronal differentiation that could be improved upon restoration of SMN expression [23,32,33]. Similarly, in our study, we observed a reduced neuronal conversion rate in only one of the two SMA iNs lines, despite having the same mutation, an identical SMN2 copy number (Table 2), and comparable levels of SMN expression to those measured via Western blot (Figure 2F and Appendix A). Notably, the reduced conversion rate is restored by increasing SMN expression via AAV9.SMN treatment. These findings suggest the existence of SMA modifiers that might act on neuronal differentiation, a topic worthy of further exploration.

Previous studies have investigated the phenotype of short neurites in SMARD1 iPSC-derived neurons, which was found to be improved with the addition of IGHMBP2 via plasmid electroporation [34,35]. We replicated these findings using iNs from SMARD1/CMT2S patients and their treatment with AAV9.IGHMBP2 (Figure 2). Similarly to the SMA lines, we observed a reduced conversion rate in one patient line with IGHMBP2 mutations. The differentiation defects have been previously described in primary motor neurons in an nmd-2J mouse model [48], but have not been reported in human lines, to our knowledge, to date. Unlike the SMA lines, treatment with AAV9.IGHMBP2 did not rescue the reduced conversion rate phenotype in our affected SMARD1/CMT2S line (Figure 2C). These findings suggest that there may be non-IGHMBP2-related pathways involved in the disease pathology, and that our in vitro iN model may help to elucidate potential disease modifiers involved in said pathway.

An additional compelling aspect of our in vitro model is its potential to evaluate VUS. In the field of diagnosis, VUS evaluation is critical in determining the significance of mutations and their impact on an individual’s health [52,53], particularly on the IGHMBP2- related disease spectrum, which exhibits variable disease courses and no phenotype-genotype correlation [10,15]. The impact of VUS on disease pathology is even more important in the case of IGHMBP2-related disorders, as there is a potential treatment on the horizon. This study presents a patient-specific method for distinguishing VUS and provides evidence of its efficacy. In particular, this study utilizes cells derived from a patient who was presented with a known pathogenic mutant allele in IGHMBP2 and a VUS at the time of diagnosis (Table 4). When converted to neurons, the VUS-iNs displayed a severe in vitro phenotype similar to that observed in previously characterized patient cells, with shorter neurites, and improvement upon treatment with AAV9.IGHMBP2 (Figure 3). In this study, this VUS was deemed to be likely pathogenic, and its negative effect could be nullified via AAV9.IGHMBP2 treatment, providing promise for future diagnostic efforts. Importantly, subsequent to this study, the VUS in question has been reclassified as pathogenic, bolstering the robustness of the in vitro model.

## 5. Conclusions

In conclusion, we have established an in vitro model of induced neurons (iNs) for the study of multiple neuromuscular disorders, mainly Spinal Muscular Atrophy and IGHMBP2-related disorders. The iNs recapitulate previous findings with iPSC derived cells, but in a faster way, while maintaining epigenetic markers that might serve as disease modifiers. The iNs are responsive to treatments already employed in the clinic and can be used as a model to test new therapeutics and evaluate disease modifiers. Moreover, we have shown that this model system is a fast and reliable way to characterize variants of uncertain significance (VUS) and determine their responsiveness to the available therapeutic interventions. In summary, our adapted protocol for the direct conversion of iNs using small molecules represents significant advancement in the study of SMA and IGHMBP2-related disorders, and we are excited to continue exploring the potential applications of these cells in future studies.

## Figures and Tables

**Figure 1 biology-12-00867-f001:**
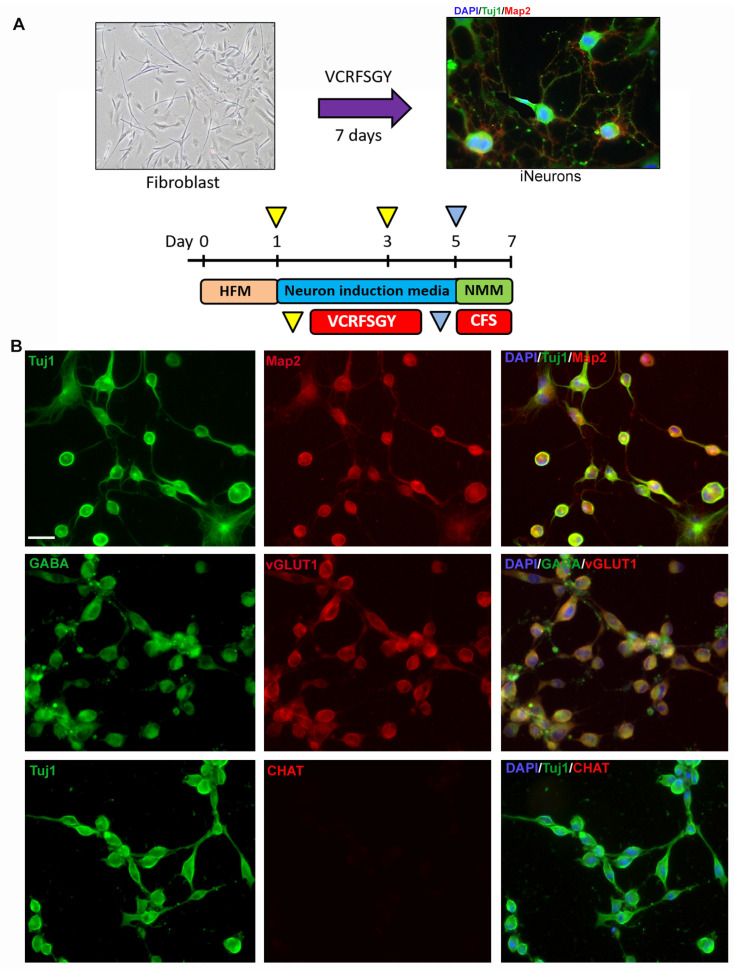
Direct conversion of skin-derived fibroblasts into induced neurons (iNs): (**A**) Schematic of the direct conversion protocol. Neurons (iNs) were generated in 7 days using seven small molecules: VPA, CHIR99021, Repsox, Forskolin, SP600125, GO6983 and Y-27632. (**B**) Fluorescent microscope imaging at 20× magnification shows directly converted iNs from a healthy line. iNs were checked for expression of pan-neuronal markers such as Tuj1 (green) and Map2 (red) (top row), as well as inhibitory neuronal marker GABA (green) and glutamatergic neuronal marker vGLUT1 (red) (middle row). The neuronal mixed population does not contain cholinergic neurons (lower row). Scale bar = 100 µm. HFM: human fibroblast medium, NMM: neuron maturation medium.

**Figure 2 biology-12-00867-f002:**
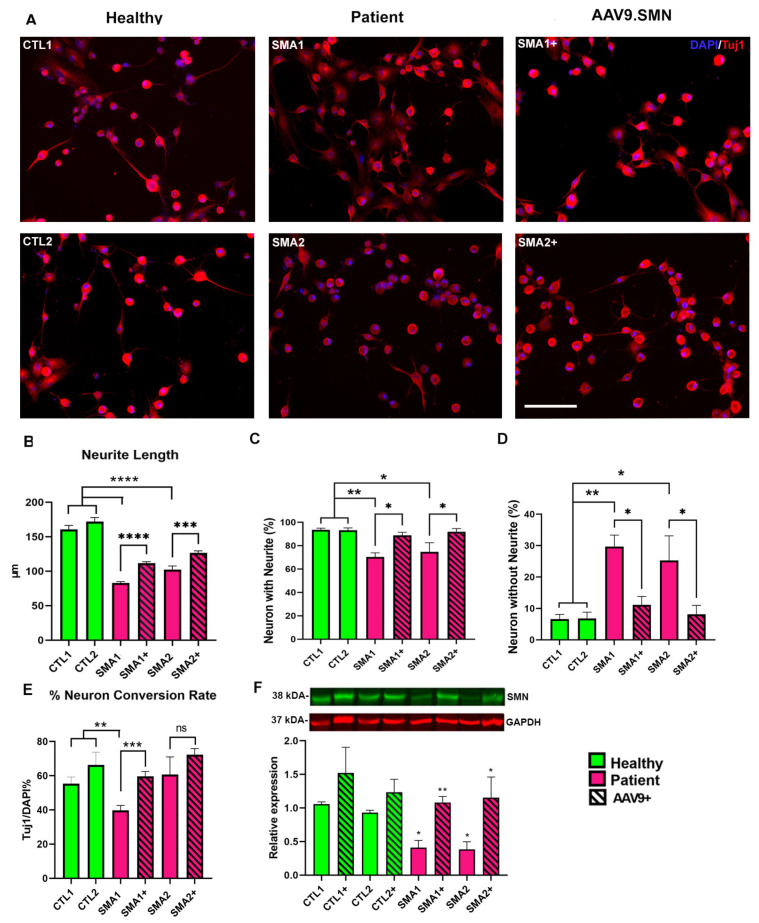
SMA induced Neurons (iNs) show aberrant morphology that is improved upon treatment with AAV9.SMN. (**A**) Representative images of Tuj1 (red) and DAPI (blue) stained iNs derived from control (left) and two SMA Type 2 patient fibroblasts (Table 2) untreated (middle) and treated with AAV9.SMN (right). iNs from both treated and untreated patients were evaluated for (**B**) neurite length on day 7, (**C**) percentage of neurons with neurites (% Tuj1-positive cells with neurites over total Tuj1-positive cells ×100) and (**D**) percentage of neurons without neurites (% Tuj1-positive cells without neurites over total Tuj1-positive cells ×100). (**E**) Additionally, the neuronal conversion rate (% Tuj1-positive cells over total DAPI-stained cells) was calculated for treated and untreated cells. (**F**) Representative Western blot showing SMN protein levels in two different control and two SMA iNs (top). Quantification of the Western blots (n = 3) normalized against GAPDH expression. Both SMA patients (SMA-1/SMA-2) showed altered iN morphology that was resolved upon treatment with AAV9.SMN. A total of 12 fields captured at 20× magnification from three differentiation experiments were analyzed using ImageJ software Version 1.53t. ANOVA, followed by Dunnett’s multiple comparison test between the mean of the controls and the mean of each line, was computed to derive the *p* value (*p*). Significance between the treated and the untreated groups was computed using unpaired *t*-tests. ns: not significant, * = *p* <0.05, ** = *p* < 0.01, *** = *p* <0.001, **** = *p* < 0.0001. Scale bar for (**A**): 100 μm.

**Figure 3 biology-12-00867-f003:**
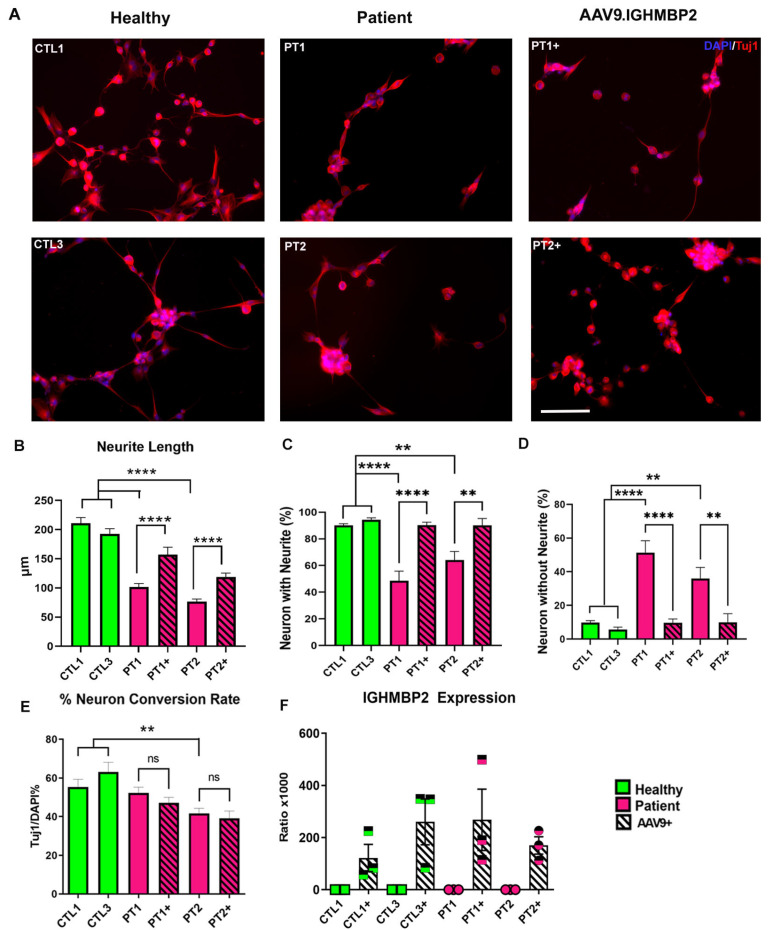
Induced Neurons from SMARD1/CMT2S patients show abnormal morphology. (**A**) Representative images of iNs derived from age-matched controls (left) and two patients on the IGHMBP2 SMA spectrum (Table 3) (middle). iNs from patients were treated with the AAV9.IGHMBP2 gene therapy which is in clinical trial (NCT05152823) (right). Resulting iN morphology was characterized using the following parameters (**B**). Neurite length on day 7, (**C**) percentage of neurons with neurites (% Tuj1-positive cells with neurites over total Tuj1-positive cells ×100) and (**D**) percentage of neurons without neurites (% Tuj1-positive cells without neurites over total Tuj1-positive cells ×100). Additionally, (**E**) the neuronal conversion rate (% Tuj1-positive cells over total DAPI-stained cells) was calculated for treated and untreated lines. (**F**) Expression of vector-derived IGHMBP2 was evaluated through ddPCR on cell lysates and standardized against YWHAZ expression. SMARD1/CMT2S iNs showed aberrant morphology that was partially corrected with induction of IGHMBP2. A total of 12 fields captured at 20× magnification from three differentiation experiments were analyzed using ImageJ software. ANOVA followed by Dunnett’s multiple comparison test between the mean of the controls and the mean of each line was computed to derive the *p* value (*p*), Significance between the treated and the untreated groups was computed using unpaired *t*-tests. ns: not significant, ** = *p* < 0.01, **** = *p* < 0.0001. Scale bar for (**A**): 100 μm.

**Figure 4 biology-12-00867-f004:**
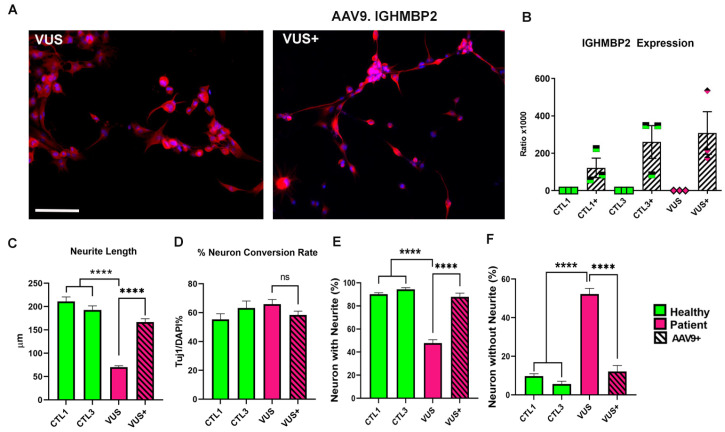
Induced Neurons from the VUS patient have the same phenotype as confirmed IGHMBP2 mutations, and show improvement upon treatment with ssAAV9.IGHMBP2. (**A**) Representative images of iN of the VUS patient (left) and with AAV9.IGHMBP2 treatment (right) (**B**) Expression of vector-derived IGHMBP2 was evaluated through ddPCR on cell lysates and standardized against YWHAZ expression. VUS iNs showed aberrant morphology on the same parameters as the confirmed SMARD1/CMT2S and improvement after treatment with the AAV9.IGHMBP2 vector. Parameters evaluated included (**C**) total neurite length on day 7, (**D**) neuronal conversion rate (% Tuj1-positive cells over total DAPI-stained cells) and other phenotypes, such as (**E**) percentage of neurons with neurites (% Tuj1-positive cells with neurites over total Tuj1-positive cells ×100), and (**F**) percentage of neurons without neurites (% Tuj1-positive cells without neurites over total Tuj1-positive cells ×100). A total of 12 fields captured at 20× magnification from three differentiation experiments were analyzed using ImageJ software. ANOVA, followed by Dunnett’s multiple comparison test between the mean of the controls and the mean of each line, was computed to derive the *p* value (*p*). Significance between the treated and the untreated groups was computed using unpaired *t*-tests., ns: not significant, **** = *p* < 0.0001. Scale bar for (**A**): 100μm.

**Table 1 biology-12-00867-t001:** SMA subtypes and classification according to onset and SMN2 copy number.

SMA Type	Onset	SMN2 Copy Number	Clinical Phenotype
Type 0	At birth	0	Most severe: death before 1 month of age
Type 1	6 months	1–2	Severe: failure to sit unaided, respiratory failure before the age of 2
Type 2	6–18 months	2–3	Intermediate: patients are able to sit but not walk unaided
Type 3	3a: before 3 years	3–4	Mild: patients are able to walk unassisted, eventually become wheelchair-bound
	3b: after 3 years		
Type 4	2nd-3rd decade of life.	4+	Mildest phenotype: normal life expectancy

**Table 2 biology-12-00867-t002:** SMA patient fibroblast lines used in this study. Fibroblasts were acquired on Coriell cell repository bank.

Patient ID	Age at Donation (Months)	Mutation	SMA Type
SMA-1	12	Homozygous for deletions of exons 7 and 8 in SMN1	Type II (3 copies of SMN2)
SMA-2	36	Homozygous deletion of exons 7 and 8 in SMN1.	Type II (3 copies of SMN2)

**Table 3 biology-12-00867-t003:** SMARD1/CMT2S patient fibroblast lines used in this study.

Patient ID	Gene	Mutation
PT1	IGHMBP2	Compound heterozygous: c.1432G > A(p.Val478Net) and c.1082T > C (p.Leu361Pro)
PT2	IGHMBP2	Compound heterozygous: c.1488C > A(pCys496 *) and C.1478C > T(p.Thr493Ile)

* makes reference to a stop codon on proteins.

**Table 4 biology-12-00867-t004:** Mutations of the uncharacterized line. Mutation 2 (c.1126G > A) was scored as VUS at the time of variant submission.

Mutations ON IGHMBP2 PATIENT VUS		
Allele 1	c.1478C > T	PATHOGENIC
Allele 2	c.1126G > A	VUS

## Data Availability

The data used in this manuscript will be made available from the corresponding authors upon reasonable request.

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
