# Peer review of "In Vitro Modeling as a Tool for Testing Therapeutics for Spinal Muscular Atrophy and IGHMBP2-Related Disorders"

_biology, 2023, doi:10.3390/biology12060867_

Round 1

Reviewer 1 Report

The manuscript by Sierra-Delgado aims to use SMA and SMARD/CMT fibroblasts directly converted into neurons to model SMA and SMARD1/CMT2S. The authors use these induced neurons to test whether AAV gene delivery can restore neurite outgrowth. SMA has been modeled extensively using iPSCs, but not with induced neurons. Using induced neurons to model SMARD1/CMT is more novel for the field, and the authors also test a variant of unknown significance. However, the characterizations of the cultures are insufficient and the overall phenotypic analyses are limited. As such, there are a number of concerns that would need to be addressed.

1. The cultures are not well characterized. What percent of neurons become GABAergic vs glutamatergic? The images in Fig 1B are not very helpful as each panel looks the same. It would be more informative to have the Tuj1 and MAP2 in the same panel and GABA and vGlut in the same panel. This would allow the reader to determine overlap/distinction between the subpopulations. Also, a nuclear dye stain needs to be included.

2. The neuron conversion rate data show clear differences between the individual fibroblast lines in all the patient backgrounds and ~50-60% differentiation efficiency. What other cell types are present in the cultures? Are they just remaining fibroblasts or do they become something else? How might those other 40-50% of cells impact the phenotype? How much variability across experiments is there within each line in terms of neuron generation? How does this conversion rate compare with other iPSC model systems and is the direct conversion still advantageous in this regard?  

3. Are astrocytes generated in these cultures? On line 48 in the abstract, it states that neurons and astrocytes were generated and infected with an AAV.

4. What is the rationale for using forebrain-like neurons instead of motor neurons or even sensory neurons (for CMT2S)?

5. With the neurite outgrowth phenotype, do the neurites ever form or do they form at the start and then die back? If the neurons were cultured for longer, would they eventually grow neurites? Is there cell death over time? Since the authors make the argument that direct conversion has advantages over iPSC-based models, it would be important to put these data in the context of what else is in the field and relate the neurite outgrowth problems to clinical observations.

6. The images in figures 2, 3, and 4 need a nuclear stain included.

7. Neurite outgrowth is restored in the three disease backgrounds with AAV delivery, but what does that mean functionally? Is there a restoration of other neuronal properties (e.g. synaptic function, axonal transport, etc?).

8. Copy editing is needed in multiple places (e.g. line 69, add a space between SMA and is; Line 72, the cytosine (C) was converted into the copyright symbol; line 100, add a space between failure and (SMARD1); Line 217, remove the ellipsis).

Reviewer 2 Report

In their manuscript Sierra-Delgado and coworkers adapt a fast direct conversion protocol to generate neuronal cells from fibroblasts obtained from SMA and SMARD patients and healthy controls. They characterized these chemically induced neurons and report differences in respect to neurite length and number between neurons derived from diseased and healthy persons that could be reversed upon viral transduction with SMN (SMA) or IGHMBP2 (SMARD) genes. They discussed this system as a fast alternative to patient-derived neuronal cells for testing of novel treatment options and categorizing genetic variant of uncertain significance. In principle, this is a straightforward approach that deserves further attention. There are, however, a number of shortcomings that should be addressed before publication.

1. The hciNs Sierra-Delgado generated in their work resemble a mixed population of GABAergic and glutamatergic neurons. Although there is evidence that glutamatergic sensory neurons contribute to SMA-like pathology in mouse models of the disease (Buettner et al. (2021) iScience 24, 103376), motor neurons are the primarily affected cells. The authors have not investigated whether cholinergic neurons were induced by their protocol. A similar protocol was originally reported by Hu et al. ((2015) Cell Stem Cell 17, 204-212) that resulted in only glutamatergic neurons but hardly any GABAergic and cholinergic ones. Due to the central importance of motor neurons for the diseases the authors looked at, the fraction of cholinergic neurons should be quantified, perhaps by using a specific marker like CHAT. Furthermore, the difference in respect to the induction of GABAergic neurons should be discussed.

2. There are protocols available for the induction of motor neuron from iPSCs based on small molecules (e.g. Bianchi et al. (2018) Stem Cell Res. doi.org/10.1016/j.scr.2018.09.006). It is not clear to this reviewer, why the authors have stuck to the protocol of Hu et al. and not taken the opportunity to take it a step further to directly induce motor neurons, a cell type with much higher relevance for SMA and SMARD. What a pity.

3. The authors use a shortened induction protocol as compared to the original work (Hu et al., 2015). Under these conditions (4 days in VCRFSGY versus 8 days), Hu et al. reported only inefficient neuron enrichment (about 1 % versus 14 %, no neuronal differentiation phase, Fig. S1C). When Hu et al. combined optimal induction and differentiation regimes, they were able to attain about 30 % of neurons in their cultures. By contrast, Sierra-Delgado et al. achieved a purity of about 50-60 % with a shortened but expectedly less efficient protocol. This is a bit difficult to believe. A possible explanation for this result may lay in the lack of specificity of their staining reactions. Comparing their Tuj1 stains with the ones performed by Hu et al., no fiber-like structures are visible within the cells but the whole cells appear to be stained. This is either unlikely to represent microtubules or due to gross overexposure. Since the authors show only single stains in their figures, although they have used double staining with DAPI for their quantification, one cannot judge the claim they made in their graphs. They need to show the double stainings their quantifications are based on and, most importantly, they need to prove the specificity of their Tuj1 labeling. They furthermore need to clarify why they get more neurons as compared to Hu et al., if they are able to prove that their staining is really specific. And they should also perform multiple stains to demonstrate/visualize the relative proportions of the different neurotransmitter types!

4. The authors restrict their analysis to morphological traits. There a many other parameters that are changed by a drop in the levels of SMN, e.g. macropinocytosis, mitochondrial status, splicing of certain mRNA precursors or expression levels of certain proteins. The authors should use some of these other parameters to characterize their chemical induction system at least in respect to the analysis of SMA on a broader basis.

5. The authors obviously are not aware that they misuse the term "conversion rate" in respect to the value they actually measure, which is rather the purity of their induced cells (the fraction of Tuj+ cells of all cells stained by DAPI, see Hu et al.)! Conversion rate would take into account the cells that were there at the beginning (the fibroblasts) and relate the number of neurons induced and differentiated by the protocol to them. The authors should use the correct term (purity) throughout the manuscript. This is important, since in the discussion the authors actually compare the purity of their cells with conversion rates as measured by Hu et al.. This is nonsense, since it means comparing apples and oranges.

Round 2

Reviewer 1 Report

Most of the previous concerns have been addressed with the revision. There are a few copy editing issues that remain (lines: 151, 254, 266, 298, 340, and the bolded text in lines 184-191. Issues include misspelling, extra period, extra spaces/lack of spaces, etc). 

With the new ICC images for the glutamatergic and GABAergic neuron generation, it seems unusual that virtually all of the neurons in the culture would simultaneously express both the excitatory and inhibitory marker. Why would this be? Reference(s) should be included to support the co-labeling overlap in these converted neurons and should be addressed in the discussion somewhere. 

Author Response

Please see the attatchment

Reviewer 2 Report

The authors improved the manuscript and responded to the points raised in the review.

Author Response

 The reviewer's invaluable guidance and expertise greatly improved the quality and clarity of our work.  We appreciate the thorough evaluation, constructive feedback, and fresh perspective provided by the reviewer. Their acceptance of the changes has strengthened the content and refined the overall structure, contributing to the excellence of the final version. We are immensely thankful for the reviewer's dedication to maintaining the publication's integrity and for their insightful and constructive feedback.